# Nonlinear Lamb Wave Micro-Crack Direction Identification in Plates with Mixed-Frequency Technique

**Liqiang Guan [1], Mingxia Zou [1], Xili Wan [1] and Yifeng Li [1,2,*]**

[1] College of Computer Science and Technology, Nanjing Tech University, Nanjing 211800, China; guanliqiangqiang@163.com (L.G.); hello_zmx@163.com (M.Z.); xiliwan@njtech.edu.cn (X.W.)

[2] Key Laboratory of Modern Acoustics of MOE, Nanjing University, Nanjing 210093, China

* Correspondence: lyffz4637@163.com

**Abstract:** This paper investigates the direction identification of micro-cracks with nonlinear components generated by Lamb wave with frequency-mixing technique. Three-dimensional finite element simulations were carried out to investigate the interaction mechanism between Lamb wave signals and micro-cracks. Upon re-visiting the conventional Lamb wave excitation signal with two kinds of fundamental frequencies ($f_1$ and $f_2$), it was found to be possible to generate new types of frequencies ($f_1 \pm f_2$) at the sideband if nonlinear sources existed in the plate. A pulse inversion method was used to extract the sideband frequency for nonlinear ultrasonic detection. By arranging piezoelectric chip arrays around the micro-crack, the acoustic nonlinearity parameter $\beta$ related to the fundamental frequency and the sideband frequency for different micro-crack directions was calibrated. It was shown that $\beta$ varied for different crack directions, which provides useful information about the scattering features of the nonlinear Lamb wave interacting with the micro-crack to characterize its directivity. Moreover, the scattering degree defined with the relative nonlinear parameter $\beta'$ of the micro-crack in different directions was investigated in detail by changing the size of the micro-crack. The outcomes showed that the forward scattering signal of the crack had a greater amplitude, whereas the backscattering signal had a smaller amplitude compared with the scattering signals in other directions from micro-cracks. In addition, the signal scattering degree in the forward direction from micro-cracks increased with the increasing micro-crack length, but decreased with increasing crack width. Furthermore, for the buried crack, the forward scattering degree of Lamb wave from micro-crack decreased as crack was buried deeper in plate. In summary, the findings of this study can help to further advance the use of nonlinear Lamb wave with the frequency-mixing technique for identifying the direction of micro-cracks.

**Keywords:** nonlinear Lamb wave; frequency mixing; pulse inversion; direction identification

## 1. Introduction

Plate-like structures are widely used in various kinds of equipment, and fatigue crack is one of their main damage forms. Fatigue cracks will cause stress concentrations in the structure. If corrective measures are not taken in time, a crack will accelerate its expansion under the action of load and eventually lead to structural fracture. Therefore, the use of non-destructive evaluation (NDE) for the detection or identification of cracks at an incipient stage is essential for maintaining the integrity and safety of a structure. Lamb waves, with their characteristics of long propagation distance and slow energy attenuation, have been considered as one of the most potential NDE monitoring technologies [1–6]. Existing linear Lamb wave monitoring technique uses changes in time-domain

signal features (e.g., acoustic velocity variation [7], mode conversion [8], time of flight (TOF) [9], and transmission and reflection coefficients [10]) to detect a small number of visible fatigue cracks, macroscopic holes, and notches. However, the technique based on the temporal signal features is insensitive to the embryonic stage of mechanical damage (much smaller than the probing wavelength) because small damage artifacts are not expected to induce evident changes in the linear features to be extracted from the ultrasonic wave.

To overcome the aforementioned deficiencies, nonlinear Lamb wave monitoring techniques based on the frequency domain characteristics of signals have attracted increasing attention in research because they can detect micro-cracks effectively. Existing research results show that Lamb wave nonlinearity comes from three main sources: the elastic nonlinear effect of structures or materials [11], the geometric nonlinearity of structures [12], and the boundary nonlinearity of contact [13]. These nonlinear components carry damage information and have been used intensively for identifying micro-cracks. Material nonlinearity is the high-order, sub-harmonic variation generated in the structural response signal under fundamental frequency excitation, which is mainly manifested in the second-order harmonic signals. Geometric nonlinearity is the harmonic spectrum that occurs symmetrically near the basic signal frequency under the excitation of the frequency modulation signal. Boundary nonlinearity is a type of local nonlinearity that can generate higher harmonic components. According to the detection principle, there are four methods for nonlinear ultrasonic detection: vibration acoustic modulation technique [14], nonlinear resonance technique [15], harmonic wave technique [16], and wave-mixing technique [17]. Vibration acoustic modulation technique has advantages in detecting interface contact state and closed cracks, but this method requires the imposition of additional low-frequency vibrations on the tested specimen and is vulnerable to boundary nonlinearity, which makes the detection system difficult to operate. The nonlinear resonance method has stringent requirements for instrument and experimental operation. The harmonic method is the most widely used method for detecting micro-cracks, but the main difficulty with harmonic generation methods is that they are greatly affected by the nonlinearity of signal transmitting/receiving equipment and inherent material nonlinearity. Wang et al. [18] used the finite element method to research the interaction between nonlinear single $S_0$ mode Lamb wave and micro-cracks embedded in a thin plate. Hong et al. [19] extended the use of temporal signal processing to Lamb wave second harmonic components and located fatigue cracks using the TOF time-domain feature of the second harmonic. Zhou et al. [20] decomposed the time reversal operator (TRO) at the second harmonic frequency and introduced an imaging algorithm based on the total focus method to locate fatigue cracks.

The signal processing capability of wave-mixing technique, which could be used to replace the nonlinear resonance method, is insensitive to the nonlinearity of the measurement system and has a multi-selective frequency mode. Nonlinear Lamb wave frequency-mixing technique relies on the interaction characteristics between two incident waves of different frequencies in the structure, thus achieving a considerable detection capability for inner defects. For an undamaged isotropic solid plate with pure elastic behaviors, no new frequency components will be generated when the two waves meet. On the contrary, in a structure that contains micro-cracks, interactions will occur in the associated regions when the two waves meet in the nonlinear region, resulting in coupling of the two waves. New frequency components will be generated in the frequency domain, including sum and difference frequencies. Nonlinear Lamb wave frequency-mixing detection has attracted increased research attention because of its great potential to detect micro-cracks and damage and has gradually been applied in NDT. Chen et al. [21] derived a set of necessary and sufficient conditions for the generation of resonance waves by two time-harmonic plane waves and obtained closed analytical solutions for the generation of resonance waves by two linearly propagating sinusoidal pulses. Lee et al. [22] conducted an exploratory study using nonlinear Lamb wave for damage detection in plates and investigated various aspects of nonlinear features by the frequency-mixing technique. Jiao et al. [23] analyzed the feasibility of the proposed acoustic nonlinear parameters and applied nonlinear Lamb wave-mixing technique to detect the degree of damage of micro-cracks in sheet metal, and the obtained

results demonstrated that the consistence between the theoretical analysis and the experimental study. However, few studies have been carried out on nonlinear Lamb wave frequency-mixing to measure the direction of micro-cracks.

This paper presents a nonlinear Lamb wave frequency-mixing technique to characterize the direction of micro-cracks by analyzing the scattering features around them. The paper is organized as follows: Section 2 introduces the basic theoretical background of nonlinear Lamb wave-mixing technique. The fundamental theory of pulse inversion and its use to extract the sideband at the sum or difference frequency signal are also described. Section 3 describes in detail the types of 3D finite-element model of micro-cracks and the process of micro-crack direction identification, and Section 4 draws conclusions.

## 2. Basic Theory of Nonlinear Lamb Waves

### 2.1. Mode Selection of Lamb Wave

Ultrasonic Lamb waves as one kind of guided waves propagating in a homogeneous and isotropic thin-plate structure [23]. When propagating in a plate structure, the ultrasound's frequency and wavenumber along the propagation direction need to satisfy the dispersion equation of Lamb waves [24]. Lamb waves propagation has dispersive and multi-mode aspects. The propagation velocity of Lamb waves is related not only to the elastic constant and material density, but also to the wave frequency and plate thickness. Figure 1 shows the dispersion curves for Lamb wave propagation in a 2-mm-thick aluminum plate.

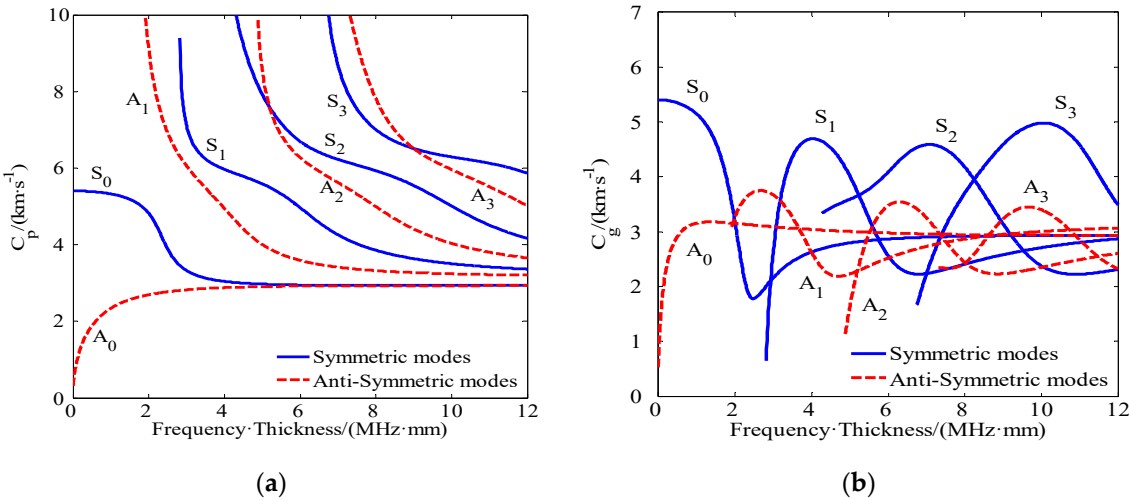

**Figure 1.** Lamb wave dispersion curves in an aluminum plate: (**a**) phase velocity; (**b**) group velocity.

The basis of defect detection in a plate is choosing the appropriate mode of Lamb wave. For a plate of given thickness, there are at least two modes of Lamb wave at a particular frequency. In practical ultrasonic testing, different Lamb wave modes have different response behaviors to defects, and this attribute will directly affect the detection results. Figure 1 shows that only two kinds of Lamb waves, $S_0$ and $A_0$, can be excited in the frequency range less than the cutoff for the A1 mode. In this frequency range, the $A_0$ mode shows more serious dispersion characteristics than the $S_0$ mode. In addition, the group velocity of the $S_0$ mode is always greater than that of the $A_0$ mode, ensuring that it will be the first wave packet to reach the receiving destination. Moreover, it is suitable for detecting micro-scale damages or micro cracks buried in the interior of a large-scale structure [13]. Hence, an appropriate excitation method should be selected to generate a single $S_0$ mode.

## 2.2. Nonlinear Lamb Wave Frequency-Mixing Excitation

The feasibility and validity of nonlinear Lamb wave to detect the micro-cracks with the change of size by frequency-mixing excitation technique were verified in theory and experiment by Professor Jiao [23]. To achieve a quantitative understanding of the inherence of the wave-mixing technique, assuming some nonlinear source such as crack damage, the input and output of the system can be simplified to the following second-order nonlinear equation [25]:

$$y(t) = \alpha x(t) + \beta x^2(t) + q(t) \tag{1}$$

where $x(t)$ represents the incident signal; $y(t)$ is the system response; $\alpha$ and $\beta$ are the first- and second-order nonlinear coefficients, respectively; and $q(t)$ is unpredictable random background noise, which is ignored in the following discussion.

In the present research, the input signal contains two frequency components, $f_1$ and $f_2$, and the incident signal $x(t)$ is assumed to take the form:

$$x(t) = A_1 \sin(2\pi f_1 t + k_1 x + \varphi_1) + A_2 \sin(2\pi f_2 t + k_2 x + \varphi_2) \tag{2}$$

where $x$ is the distance of wave propagation and the pairs $(A_1, A_2)$, $(f_1, f_2)$, $(k_1, k_2)$, and $(\varphi_1, \varphi_2)$ are respectively the sinusoidal signal amplitudes, frequencies, wave numbers, and phases of the two sinusoidal components. Then the corresponding system response output $y(t)$ can be determined by substituting Equation (2) into Equation (1), which gives:

$$\begin{aligned}
y(t) = \ & \alpha A_1 \sin(2\pi f_1 t + k_1 x + \varphi_1) + \alpha A_2 \sin(2\pi f_2 t + k_2 x + \varphi_2) - \\
& \beta \frac{A_1^2}{2} \cos[2\pi(2f_1)t + 2k_1 x + \varphi_1] - \beta \frac{A_2^2}{2} \cos[2\pi(2f_2)t + 2k_2 x + \varphi_2] + \\
& \beta A_1 A_2 \cos[2\pi(f_2 - f_1)t + (k_2 - k_1)x + (\varphi_2 - \varphi_1)] - \\
& \beta A_1 A_2 \cos[2\pi(f_2 + f_1)t + (k_2 + k_1)x + (\varphi_2 + \varphi_1)]
\end{aligned} \tag{3}$$

Using the Fourier transformation, the response signal can be switched from the time domain to the frequency domain and the fundamental wave, second harmonic, sum and difference frequencies can be separated. Therefore, the response signal becomes:

$$\begin{aligned}
Y(f) = \ & \alpha A_1 \frac{j}{2}(\delta(f + f_1)e^{-j(\varphi_1 + k_1 x)} - \delta(f - f_1)e^{j(\varphi_1 + k_1 x)}) + \\
& \alpha A_2 \frac{j}{2}(\delta(f + f_2)e^{-j(\varphi_2 + k_2 x)} - \delta(f - f_2)e^{j(\varphi_2 + k_2 x)}) - \\
& \beta \frac{A_1^2}{4}[\delta(f + 2f_1)e^{-j2(\varphi_1 + k_1 x)} + \delta(f - 2f_1)e^{j2(\varphi_1 + k_1 x)}] - \\
& \beta \frac{A_2^2}{4}[\delta(f + 2f_2)e^{-j2(\varphi_2 + k_2 x)} + \delta(f - 2f_2)e^{j2(\varphi_2 + k_2 x)}] + \\
& \beta \frac{A_1 A_2}{2}[\delta(f + f_2 - f_1)e^{j((k_2 - k_1)x + (\varphi_2 - \varphi_1))} + \delta(f - f_2 + f_1)e^{-j((k_2 - k_1)x + (\varphi_2 - \varphi_1))}] - \\
& \beta \frac{A_1 A_2}{2}[\delta(f + f_2 + f_1)e^{j((k_2 + k_1)x + (\varphi_2 + \varphi_1))} + \delta(f - f_2 - f_1)e^{j((k_2 + k_1)x + (\varphi_2 + \varphi_1))}].
\end{aligned} \tag{4}$$

Equation (4) described the nonlinear system effects for mixed-frequency excitation. Therefore, under mixed-frequency excitation, the existence of nonlinear sources such as structural damage could be judged by analyzing whether there were sum and difference frequency components in the spectrum of the output signal.

When the experimental condition of the propagation distance and the wavenumber is identical, the second-order acoustic nonlinear coefficient $\beta$ can be simplified to [23,26]:

$$\beta = \frac{A_{f_1 - f_2}}{A_1 A_2} \tag{5}$$

where $A_{f_1 - f_2}$ is the amplitude of the difference frequency in the spectrum of the response signal. The relative parameter $\beta$ contains the essential nonlinear properties of wave propagation. It can therefore serve as a primary index to be monitored for quantitative characterization of micro-damage.

### 2.3. Pulse Inversion Method of Frequency-Mixing Excitation

Pulse inversion technique was originally used for extracting the second harmonic in medical field [27], which can effectively extract second harmonic time-domain signal, enhance the amplitude of second harmonic signal, and suppress odd harmonic components produced mainly by the experimental system [28].The method is used for further processing of the output mixing-wave signal in Equation (3). When the initial phase of the exciting ultrasonic signal is 0°, the output of the nonlinear response is as follows:

$$
\begin{aligned}
y_1(t) = \quad & \alpha A_1 \sin(2\pi f_1 t + k_1 x) + \alpha A_2 \sin(2\pi f_2 t + k_2 x) - \\
& \beta \tfrac{A_1^2}{2} \cos[2\pi(2f_1)t + 2k_1 x] - \beta \tfrac{A_2^2}{2} \cos[2\pi(2f_2)t + 2k_1 x] + \\
& \beta A_1 A_2 \cos[2\pi(f_2 - f_1)t + (k_2 - k_1)x] - \beta A_1 A_2 \cos[2\pi(f_2 + f_1)t + (k_2 + k_1)x].
\end{aligned}
\tag{6}
$$

If the initial phase of the exciting ultrasonic signal is 180°, the output of the nonlinear response becomes:

$$
\begin{aligned}
y_2(t) = \quad & -\alpha A_1 \sin(2\pi f_1 t + k_1 x) - \alpha A_2 \sin(2\pi f_2 t + k_2 x) - \\
& \beta \tfrac{A_1^2}{2} \cos[2\pi(2f_1)t + 2k_1 x] - \beta \tfrac{A_2^2}{2} \cos[2\pi(2f_2)t + 2k_2 x] + \\
& \beta A_1 A_2 \cos[2\pi(f_2 - f_1)t + (k_2 - k_1)x] - \beta A_1 A_2 \cos[2\pi(f_2 + f_1)t + (k_2 + k_1)x].
\end{aligned}
\tag{7}
$$

Adding the response signals $y_1(t)$ and $y_2(t)$ yields:

$$
\begin{aligned}
y_3(t) = \quad & -\beta A_1^2 \cos[2\pi(2f_1)t + 2k_1 x] - \beta A_2^2 \cos[2\pi(2f_2)t + 2k_2 x] + \\
& 2\beta A_1 A_2 \cos[2\pi(f_2 - f_1)t + (k_2 - k_1)x] - 2\beta A_1 A_2 \cos[2\pi(f_2 + f_1)t + (k_2 + k_1)x].
\end{aligned}
\tag{8}
$$

and the Fourier transform converts Equation (8) into:

$$
\begin{aligned}
Y_3(f) = \quad & -\beta A_1^2[\delta(f + 2f_1) + \delta(f - 2f_1)]e^{j2k_1 x} - \beta A_2^2[\delta(f + 2f_2) + \delta(f - 2f_2)]e^{j2k_2 x} \\
& +2\beta A_1 A_2[\delta(f + f_2 - f_1) + \delta(f - f_2 + f_1)]e^{j(k_2 - k_1)x} - \\
& 2\beta A_1 A_2[\delta(f + f_2 + f_1) + \delta(f - f_2 - f_1)]e^{j(k_2 + k_1)x}.
\end{aligned}
\tag{9}
$$

Equation (9) reveals that the fundamental frequency component of the original response signal is canceled out in the signal processed by pulse inversion, whereas the amplitudes of the corresponding second-order, difference frequency, and sum frequency components were increased. In other words, after pulse inversion processing, the fundamental frequency is effectively eliminated, and the amplitudes of the difference frequency and sum frequency components in the response signal become twice as large as that of the original response signal.

## 3. Three-Dimensional Finite-Element Simulation

### 3.1. 3D Models of an Aluminum Plate

In this study, two types of three-dimensional finite-element simulation models had been constructed using the Abaqus software, and the Abaqus/Explicit analysis step was employed to simulate Lamb wave propagation in plate.

#### 3.1.1. Through-Thickness Micro-Crack Model

The research object of the model was an aluminum plate. The size and parameters of the aluminum plate are shown in Table 1.

**Table 1.** Model dimensions and aluminum plate material parameters.

| Dimensions (Length × Width × Height) | Density $\rho$ | Elasticity Modulus $E$ | Poisson's Ratio $\lambda$ |
| --- | --- | --- | --- |
| 300 × 300 × 2 mm | 2700 kg/m$^3$ | 70 GPa | 0.33 |

For convenience in discussion, a coordinate system was defined with its origin at the center of the plate, as shown in Figure 2a. Two piezoelectric chips ($P_{t1}$ and $P_{t2}$) were used to excite the incident Lamb wave signal; they were located at (−70 mm, 0 mm) and were symmetrically distributed on the upper and lower surfaces of the plates. Another eight piezoelectric chips were bound to the upper surface of the aluminum plate to collect Lamb wave structural response signals, which were labeled as $P_{r1}$–$P_{r8}$, forming an 80 mm diameter circular transducer network. These eight piezoelectric chips were located at $r$ = 40 mm and $\theta$ = 0° ($P_{r5}$), 45° ($P_{r6}$), 90° ($P_{r7}$), 135° ($P_{r8}$), 180° ($P_{r1}$), 225° ($P_{r2}$), 270° ($P_{r3}$), and 315° ($P_{r4}$) [29]. The through-thickness micro-crack was modeled by an extruded entity, with the in-plane midpoint of the micro-crack at (0 mm, 0 mm) and the shape of the 3D micro-crack being an elliptical cylinder. The crack profile was elliptical, and the major and minor axes of the ellipse determined the length ($l$) and width ($w$) of the micro-crack. The height ($h$) of the micro-crack was perpendicular to the upper and lower surfaces of the plate as shown in the enlarged micro-crack in damaged part in Figure 3. It was located at the center of the aluminum plate in the $y$-axis direction. The orientation of the micro-crack is the direction of micro-crack expanding, in the models the crack direction is along the micro-crack's length $l$, that is, the $y$-axis direction. Three-dimensional eight-node brick elements in Abaqus (C3D8R) were used to mesh the plane. To ensure the computational efficiency and accuracy of the model, it was essential to adopt the appropriate grid cell edge length ($L_e$) and the integration time step ($\Delta t$) calculated by Equations (10) and (11) [16]. For sidebands at the sum frequency (1200 kHz), the calculated mesh element size was 0.259 mm, and the time step was 41.667 ns. Because the model thickness was 2 mm, the grid cell edge length was selected as 0.2 mm to ensure that the grid along the direction of plate thickness was divided into an integer number of cells. A time step of 40 ns was deemed sufficient to ensure accuracy [30]:

$$L_e \leq \frac{\lambda_{min}}{10} \tag{10}$$

$$\Delta t \leq \frac{1}{20 f_{max}}. \tag{11}$$

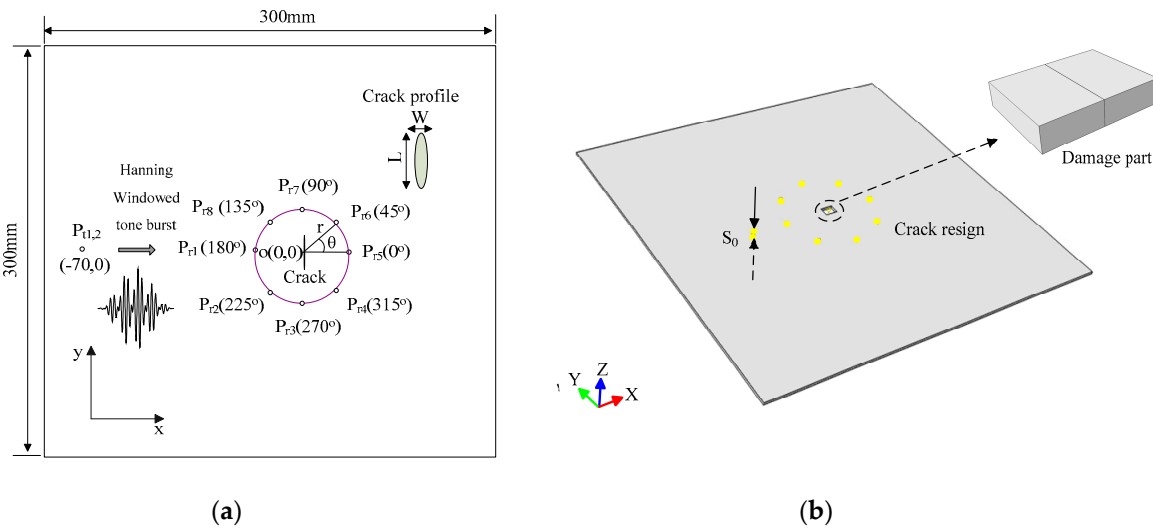

(**a**)                    (**b**)

**Figure 2.** Damage model of aluminum plate: (**a**) schematic diagram; (**b**) three-dimensional (3D) model.

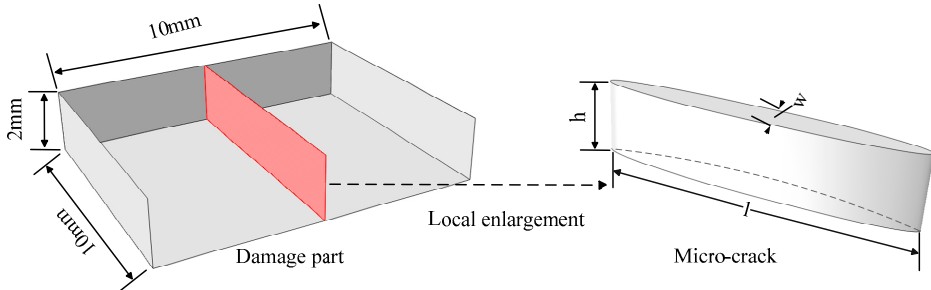

**Figure 3.** Schematic diagram of profiles of a three-dimensional (3D) through-thickness micro-crack.

According to the results of the analysis in Section 2, an exciting single $S_0$ mode Lamb wave signal could facilitate the extraction and analysis of the response signal. To obtain a single $S_0$ mode, two piezoelectric chips ($P_{t1}$ and $P_{t2}$) were used to excite the Lamb wave satisfying the following conditions: (i) $P_{t1}$ and $P_{t2}$ vertically applied stress at the same time on the upper and lower surfaces of the plate; (ii) $P_{t1}$ and $P_{t2}$ applied stress in opposite directions, as shown in Figure 2b.

### 3.1.2. Buried Micro-Crack Model

Referring to the schematic of Figure 2, the same configuration was used in this section to simulate another form of buried micro-crack. The micro-crack type was delamination defect and it was modeled by embedding the cohesive element into an entity structure as shown in Figure 4. The cohesive element and entity surface had duplicate node numbers at the same location. When the applied force reaches the critical value of the strength of the element, the cohesive element from damage to final failure could represent the delamination failure mode of the structure. Hard normal contact and friction tangential contact were applied to the interface between the cohesive element and the entity to prevent nodes penetration. Such method to modulate a seam crack with a zero initial clearance between the two surfaces of entity, so the main difference between the seam crack and the crack with an elliptical profile was that the width of the embedded seam crack was zero in the case of without applied loads. As shown in Figure 4, the delamination crack width ($w$) was zero, the length ($l$) and height ($h$) of micro-crack were parallel and vertical to the upper and lower surfaces of the plate, respectively. For buried micro-crack situation, the height ($h$) of micro-crack was less than the thickness of the plate and the micro-crack without free surface. In this case, the center of the crack coincides with the center of the plate, and the formula to quantitatively express the depth of crack burial could be defined as $(d - h)/2$.

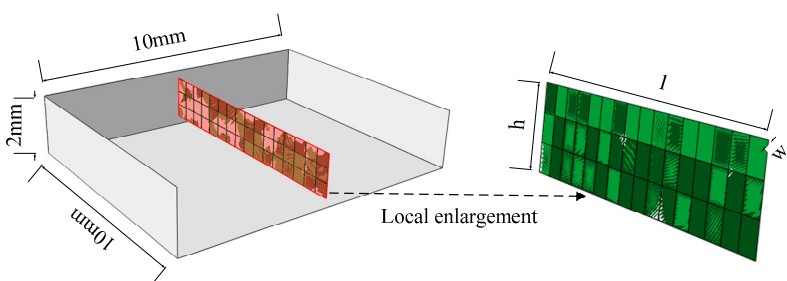

**Figure 4.** Schematic diagram of profiles of a three-dimensional (3D) buried micro-crack.

### 3.2. Simulation Results and Analyses

### 3.2.1. Signal Excitation and Processing

To investigate the nonlinear Lamb wave scattering features for different micro-crack in different directions, the selection and processing of the excitation signal is crucial. The excitation signal is a sinusoidal mixed-frequency signal modulated by the Hanning window:

$$x(t) = [H(t) - H(t - \frac{N}{f_c})](1 - \cos(\frac{2\pi f_c t}{n}))(\sin(2\pi f_1 t + \varphi_1) + \sin(2\pi f_2 t + \varphi_2)) \tag{12}$$

where H($t$) is the Heaviside step function; $f_c$ is the center frequency of the excitation signal, which is equal to the half the sum of the fundamental frequency components ($f_1$ and $f_2$); $\varphi_1$ and $\varphi_2$ are the phases of the incident frequency-mixing signal, respectively; and $N$ is the number of wave peaks of the modulated sine signal. The number of cycles for the excitation signals is 22, which ensures that the generated sum and difference frequencies will not overlap with the excitation frequency components in the frequency domain. The time-domain waveform of the excitation signal is shown in Figure 5a. The incident signals with 0° and 180° initial phase are used to excite respectively. The corresponding spectrum is shown in Figure 5b.

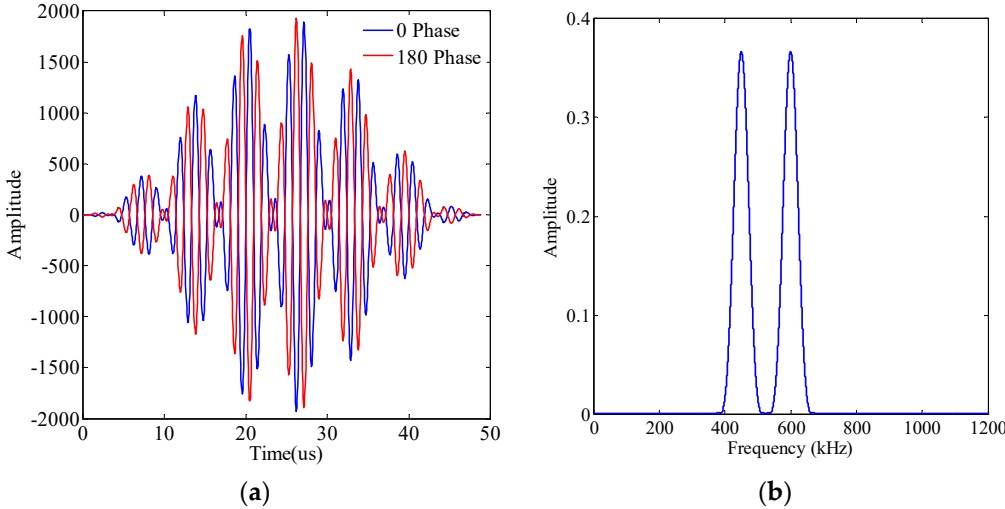

(**a**)　　　　　　　　　　　　　　　　　(**b**)

**Figure 5.** Excitation signal in the aluminum plate: (**a**) time-domain signal; (**b**) frequency-domain spectrum.

The numerical simulation was carried out according to the simulated schematic diagram of the damaged aluminum plate shown in Figure 3. Two actuators $P_{t1}$ and $P_{t2}$ (labeled as $P_{t1,2}$) were coupled to excite the $S_0$ mode Lamb wave, and then eight sensors $P_{r1}$–$P_{r8}$ received the structural response signals in the form of stress. The received signal collected by $P_{r5}$ (0°) at the micro-crack ($l$ = 6 mm, $w$ = 1 µm) is shown in Figure 6a. Two out-of-phase damage signals are displayed because of phases of the excitation signal were 0° and 180° respectively. In the spectrum of the response signal, a relatively obvious difference frequency (150 kHz) component was produced in the sideband of the basic frequencies ($f_1$ = 450 kHz, $f_2$ = 600 kHz), whereas the sum frequency (1050 kHz) component with small amplitude was barely seen in Figure 6b.

According to the above analyses, when incident signals with phase 0° and 180° were applied to the damaged plate, the corresponding structural response signals were out of phase. On the basis of pulse inversion theory, the collected damage signals at different phases are superposed in the time domain, as shown in Figure 7a and the corresponding frequency spectrum is shown in Figure 7b. In the inset of Figure 7c, the initial damage signal and the superposed signal in the frequency domain are given for comparison; after pulse inversion, the amplitude of the difference frequency increased,

but the amplitude of the fundamental frequency signals at 450 kHz and 600 kHz were eliminated. Therefore, in the nonlinear ultrasonic detection without filter module, the pulse inversion method can be used to filter the ultrasonic signal to enhance the amplitude of the difference frequency signal and eliminate the fundamental frequency signal.

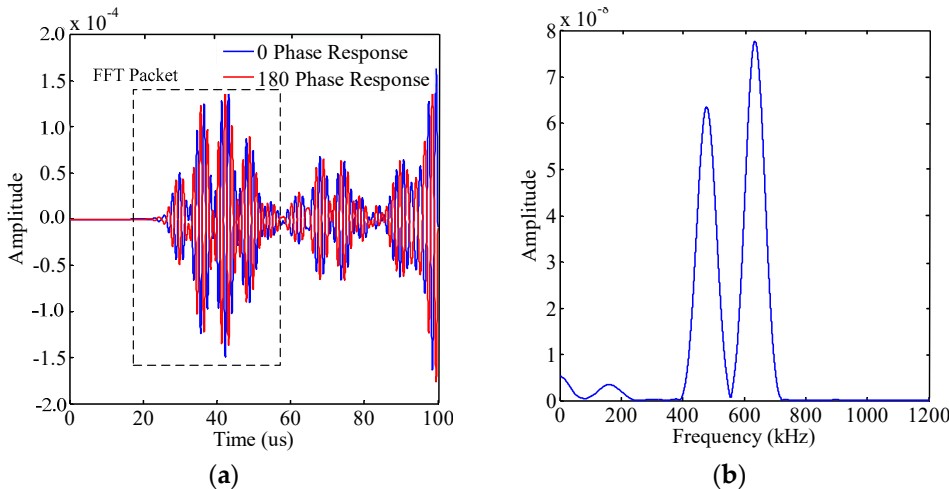

**Figure 6.** Damage signals collected from sensing path ($P_{t1,2}$–$P_{r5}$): (**a**) time-domain signal; (**b**) frequency-domain spectrum.

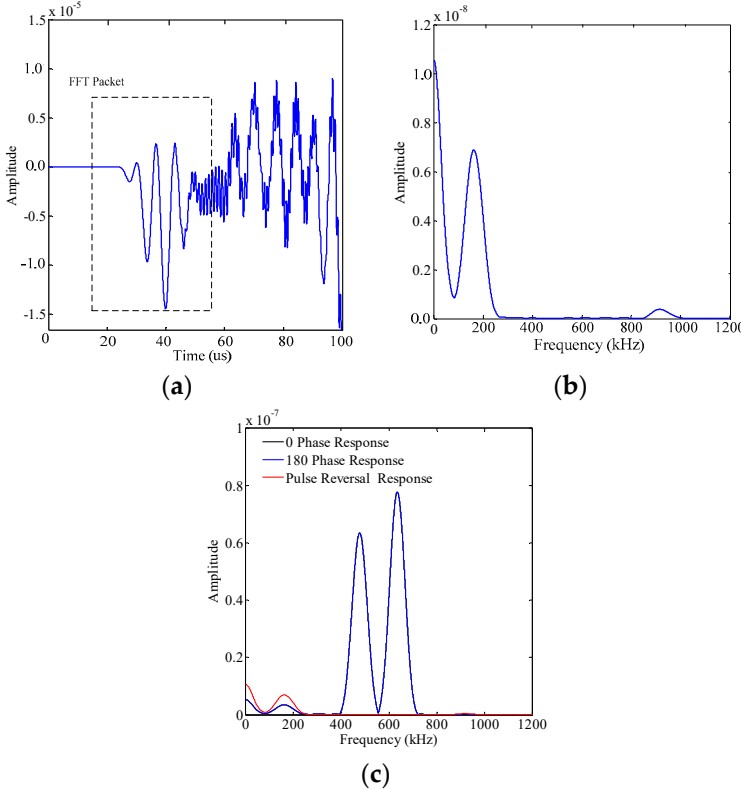

**Figure 7.** Superposed response signals after pulse inversion: (**a**) time-domain signal; (**b**) frequency-domain spectrum; (**c**) frequency-domain comparison.

### 3.2.2. Through-Thickness Crack Simulation Results and Analyses

To investigate the scattering features of nonlinear Lamb waves around a micro-crack to characterize its directivity, two groups of simulations were performed for the 3D aluminum plate with various

micro-crack lengths and widths. The simulations of the first group were performed on through-thickness crack model with fixed width ($w$ = 1 μm) and various lengths ($l$ = 2, 4, 6, 8 mm). Figure 8 shows the directivity patterns of the acoustic nonlinear coefficient $\beta$ for different micro-crack directions, when the mixing frequencies are 450 kHz and 600 kHz. From Figure 8a, the acoustic nonlinear coefficient $\beta$ at $\theta = 0°$ is slightly smaller than $\beta$ at $\theta = 45°$ when the length of the micro-crack is 2 mm. The acoustic nonlinear coefficient $\beta$ is larger at $\theta = 0°$ than in other directions when the micro-crack length increases to 4, 6, and 8 mm, as shown in Figure 8b–d. Moreover, the amplitude of the nonlinear parameter $\beta$ is the smallest at $\theta = 180°$ than in other directions when the micro-crack length varies from 2 mm to 8 mm. In general, the directivity patterns show that the amplitude of $\beta$ in the forward scattering signal of the crack was the largest, whereas the backscattering signal had a smaller amplitude than the scattering signal for other micro-crack directions. Therefore, the property of the largest forward-scattering and smallest backward-scattering features could be used to characterize micro-crack direction. The micro-crack was perpendicular to the connecting line between the largest forward-scattering point collected by $P_{r5}$ (0°) and the smallest backward-scattering point collected by $P_{r1}$(180°). Moreover, with increasing micro-crack length, the curves of the directivity patterns become more acute, which means that the directivity of long micro-cracks becomes more obvious and the direction of long micro-cracks is easier to identify.

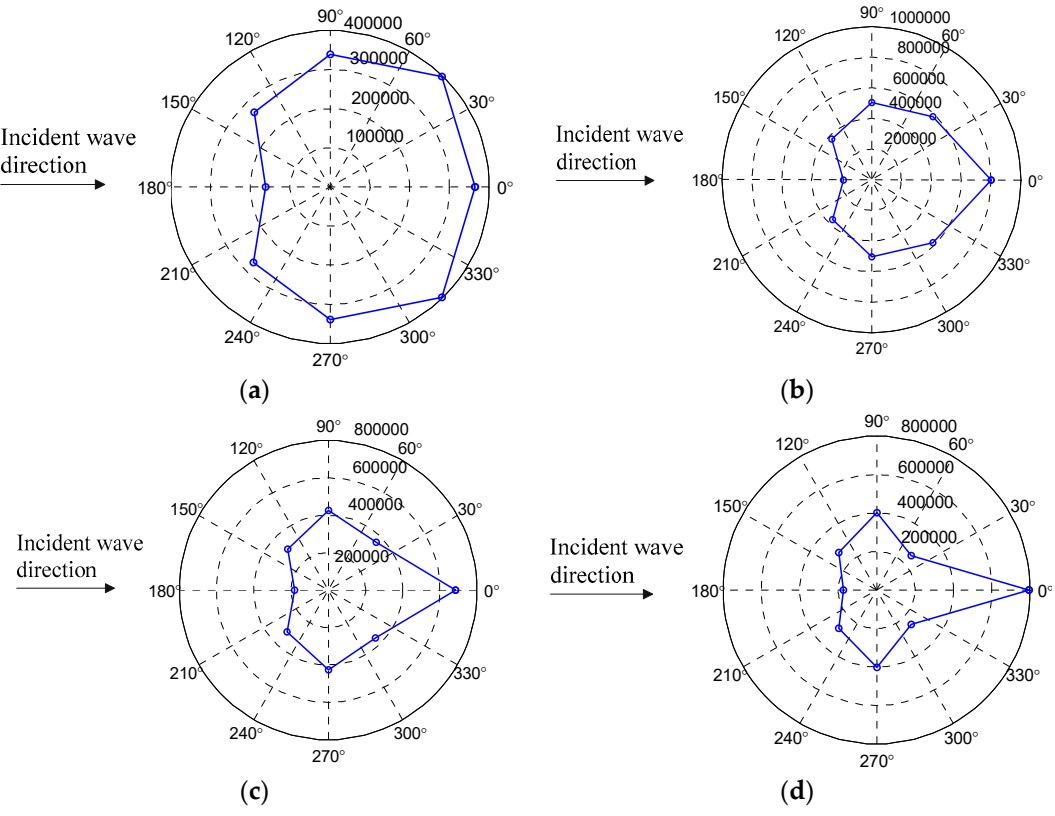

**Figure 8.** Directivity patterns of nonlinear parameter $\beta$ for (**a**) 2 mm; (**b**) 4 mm; (**c**) 6 mm; (**d**) 8 mm length micro-cracks with fixed 1 μm width.

The simulations of the second group were performed on through-thickness crack model with a fixed length ($l$ = 8 mm) and various widths ($w$ = 1, 5, 8, 10 μm) at the excited mixing frequencies $f_1$ = 450 kHz and $f_2$ = 600 kHz. The nonlinear parameters $\beta$ for different micro-crack directions were invested, and the outcomes are shown in Figure 9. The amplitude of the nonlinear parameter $\beta$ at $\theta = 0°$ was larger than in other directions, and the smallest value was obtained at $\theta = 180°$. And the figure reflects similar nonlinear characteristics, that the amplitude of the nonlinear parameter $\beta$ was the largest in the forward-scattering direction and showed the smallest value in the backward scattering

direction. Moreover, with decreasing micro-crack width, the curves of the directivity patterns became more acute and the forward scattering point gradually approaches the boundary of the circle, which means that the directivity of narrow micro-cracks became more obvious and that the direction of narrow micro-cracks was easier to identify.

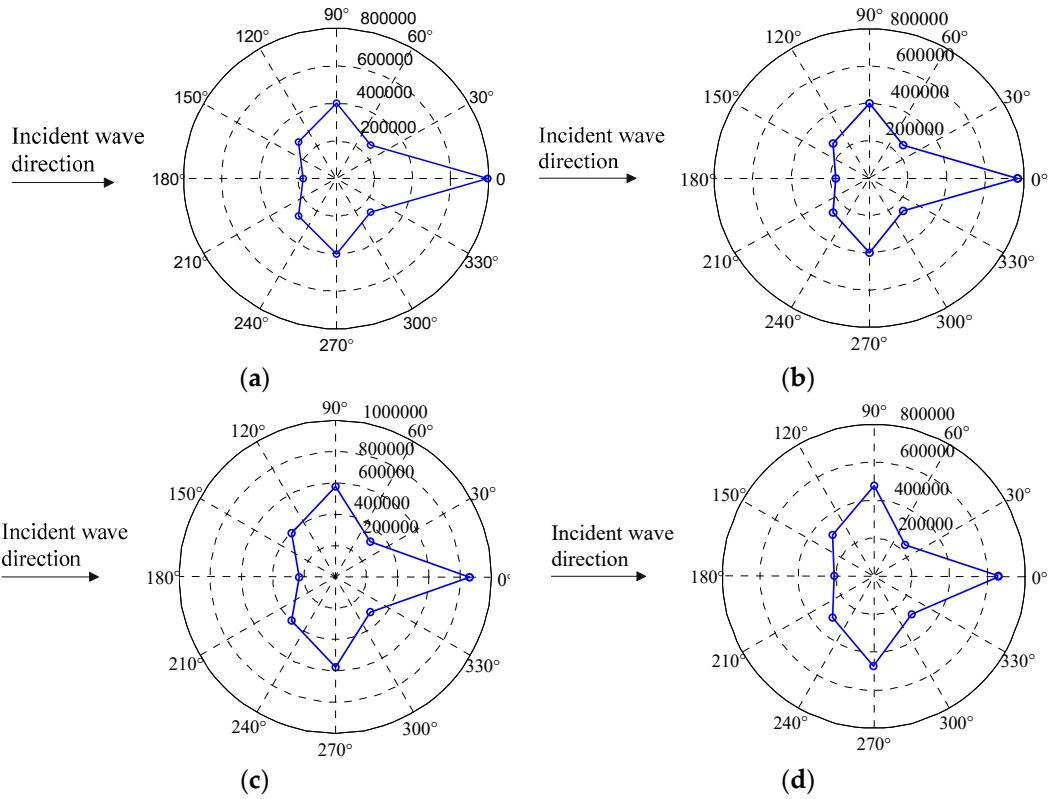

**Figure 9.** Directivity patterns of nonlinear parameter $\beta$ for (**a**) 1 μm; (**b**) 5 μm; (**c**) 8 μm; (**d**) 10 μm width micro-cracks with fixed 8 mm length.

To illustrate further the relationship between acoustic nonlinearity and crack length and width, three kinds of frequency combinations were selected as the incident signal, as shown in Table 2.

**Table 2.** Types of mixed-frequency signals.

| Sequence Number (i) | Cycle (n) | $f_1$ | $f_2$ | $f_1 + f_2$ | $f_2 - f_1$ |
|---|---|---|---|---|---|
| 1 | 22 | 450 kHz | 600 kHz | 1050 kHz | 150 kHz |
| 2 | 22 | 470 kHz | 650 kHz | 1120 kHz | 180 kHz |
| 3 | 22 | 500 kHz | 700 kHz | 1200 kHz | 200 kHz |

Damage signals for different mixed-frequency combinations were collected. To reflect intuitively the directivity of the scattering signals, the relative acoustic nonlinearity $\beta'$ is presented and defined as:

$$\beta' = \frac{\beta_{r_5}}{\beta_{r_6}} \tag{13}$$

where $\beta_{r_5}$ and $\beta_{r_6}$ represent the acoustic nonlinear coefficient $\beta$ of the damage signal as collected by $P_{r5}$ and $P_{r6}$, respectively. The acoustic relative nonlinearity $\beta'$ can reflect the directivity extent of the scattering signals and can be used for a quantitative evaluation of forward scattering. With the increasing amplitude of $\beta'$, the directivity extent of the scattering signals will become more obvious.

The relative variations of the acoustic nonlinearity parameter $\beta'$ with micro-crack lengths ($l = 2, 4, 6, 8$ mm) at fixed width ($w = 1$ μm) for three kinds of excited mixed-frequency signal sources

are displayed in Figure 10a. The magnitude of $\beta'$ increased continuously with increasing micro-crack length at the mixing frequencies ($f_1$ = 450 kHz, $f_2$ = 600 kHz). As the center frequency of the excitation signal increased, the magnitude of $\beta'$ first increased and then decreased with increasing micro-crack length at both mixing frequencies ($f_1$ = 470 kHz, $f_2$ = 650 kHz) and ($f_1$ = 500 kHz, $f_2$ = 700 kHz), which means that the forward scattering increased with increasing micro-crack length at the initial stage of micro-crack and then decreased with increasing micro-crack length. A straight explanation is that as the micro-crack became longer, the contact stiffness of the interface increased first and then decreased. From the theory of the contact acoustic nonlinearity, as the contact stiffness of the interface decreased, that is, as the micro-crack became longer, the acoustic nonlinearity increased [31]. Moreover, the variation of the relative acoustic nonlinearity parameter $\beta'$ for micro-cracks of fixed length ($l$ = 8 mm) and various widths ($w$ = 1, 5, 8, 10 μm) is shown in Figure 10b. $\beta'$ monotonically decreased with increasing micro-crack width; in the case of three types of mixing frequencies, indications were apparent of a decreasing trend for the forward-scattering degree with increasing micro-crack width. As the micro-cracks become wider, the gap between its two interfaces becomes larger, consequently, parts of the two interfaces are not in contact during the compressional phase of the incident waves, and thus the contact area is reduced. In addition, when the micro-crack was small enough, the scattering features of the nonlinear Lamb wave interacting with the micro-crack began to become inconspicuous. This indicated that the relative nonlinearity coefficient $\beta'$ was less than one as shown in the left lower corner of Figure 10a, which means that the scattering signals no longer had directivity. In order to satisfy the monotonic changing trend of the acoustic nonlinearity parameter $\beta'$, the frequency combination of 450 kHz and 600 kHz should be selected in practice.

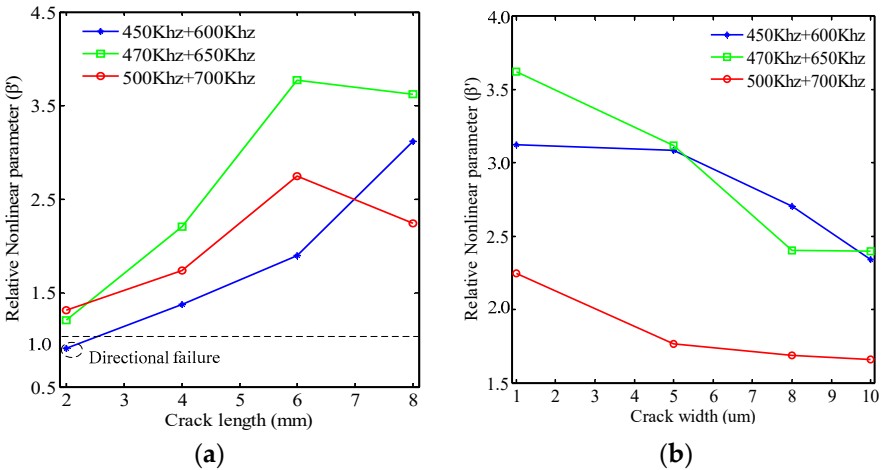

**Figure 10.** Relative nonlinear parameter $\beta'$ for (**a**) crack-length variation; (**b**) crack-width variation with three types of mixed-frequency incident signals.

### 3.2.3. Buried Crack Model Simulation Results and Analyses

To confirm the validity of nonlinear Lamb wave-mixing technique in identifying the direction of micro-crack, the acoustic nonlinear coefficient $\beta$ for buried crack was investigated. The simulations of the first group were performed on buried crack model with fixed height ($h$ = 1.8 mm) and various lengths ($l$ = 2, 4, 6, 8 mm) when the mixing frequencies are 450 kHz and 600 kHz and the results are shown in Figure 11. The results reflected the similar nonlinear characteristics as for the through-thickness crack that the amplitude of $\beta$ in the forward-scattering direction of the crack was the largest, whereas the backscattering signal had a smaller amplitude than the scattering signal of the other micro-crack directions. The direction pattern of nonlinear parameter show that the micro-crack was perpendicular to the connecting line between the largest forward-scattering point collected by $P_{r5}$ (0°) and the smallest backward-scattering point collected by $P_{r1}$(180°) and with increasing micro-crack length, the

curves of the directivity patterns become more acute, which means that the directivity of long buried micro-cracks was more obvious and easier to identify like the through-thickness crack.

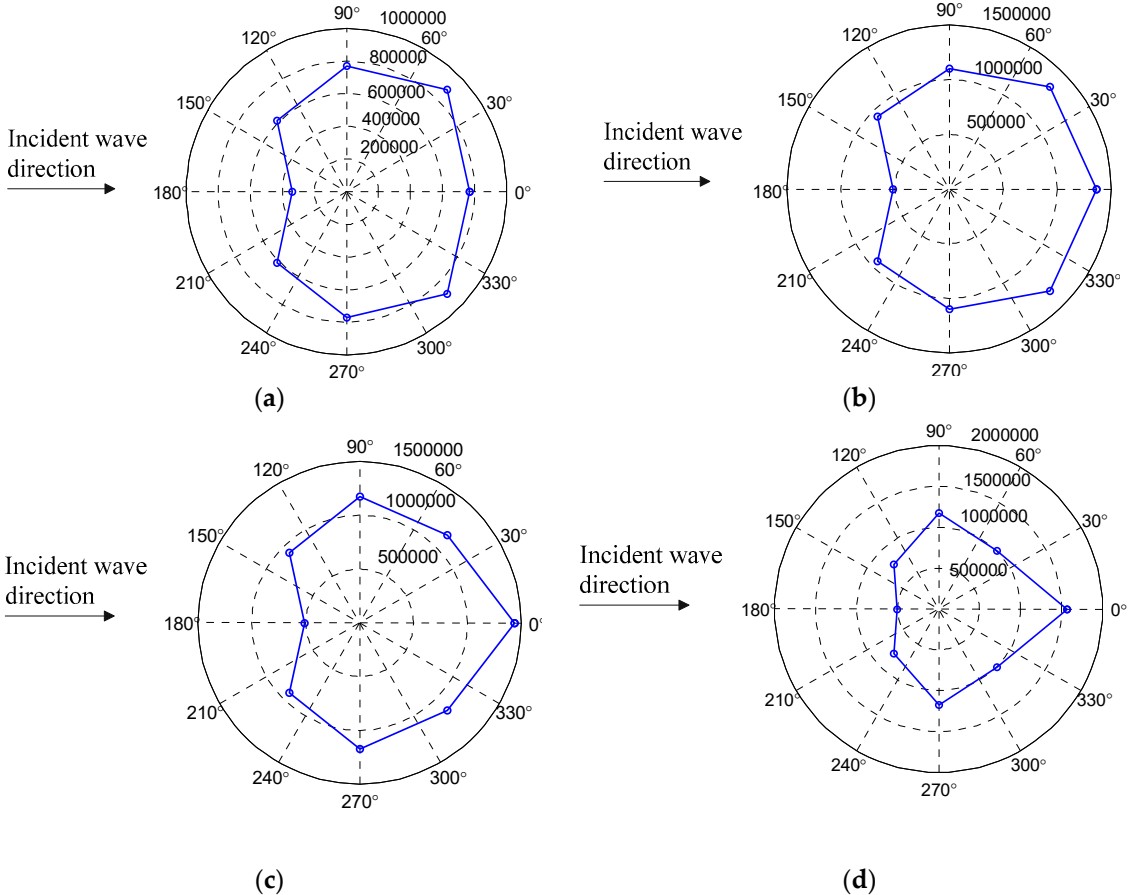

**Figure 11.** Direction patterns of nonlinear parameter $\beta$ for (**a**) 2 mm; (**b**) 4 mm; (**c**) 6 mm; (**d**) 8 mm length micro-cracks with fixed 1.8 mm height.

For buried micro-crack, the simulations of the second group were performed on model with fixed length ($l$ = 8 mm) and various heights ($h$ = 1.4, 1.6, 1.8, 2.0 µm) at the excited mixing frequencies $f_1$ = 450 kHz and $f_2$ = 600 kHz. The results of nonlinear parameter $\beta$ for different micro-crack dimensions are displayed in Figure 12 and which show that the amplitude of the nonlinear parameter $\beta$ at $\theta$ = 0° was larger than in other directions and the smallest value was obtained at $\theta$ = 180°. The direction patterns of nonlinear parameter $\beta$ mean that the largest and smallest values were obtained in the forward and backward scattering directions, respectively. Moreover, with increasing micro-crack height, the curves of the directivity patterns became more acute, which means that the directivity of higher buried micro-cracks became more obvious and its direction was easier to identify.

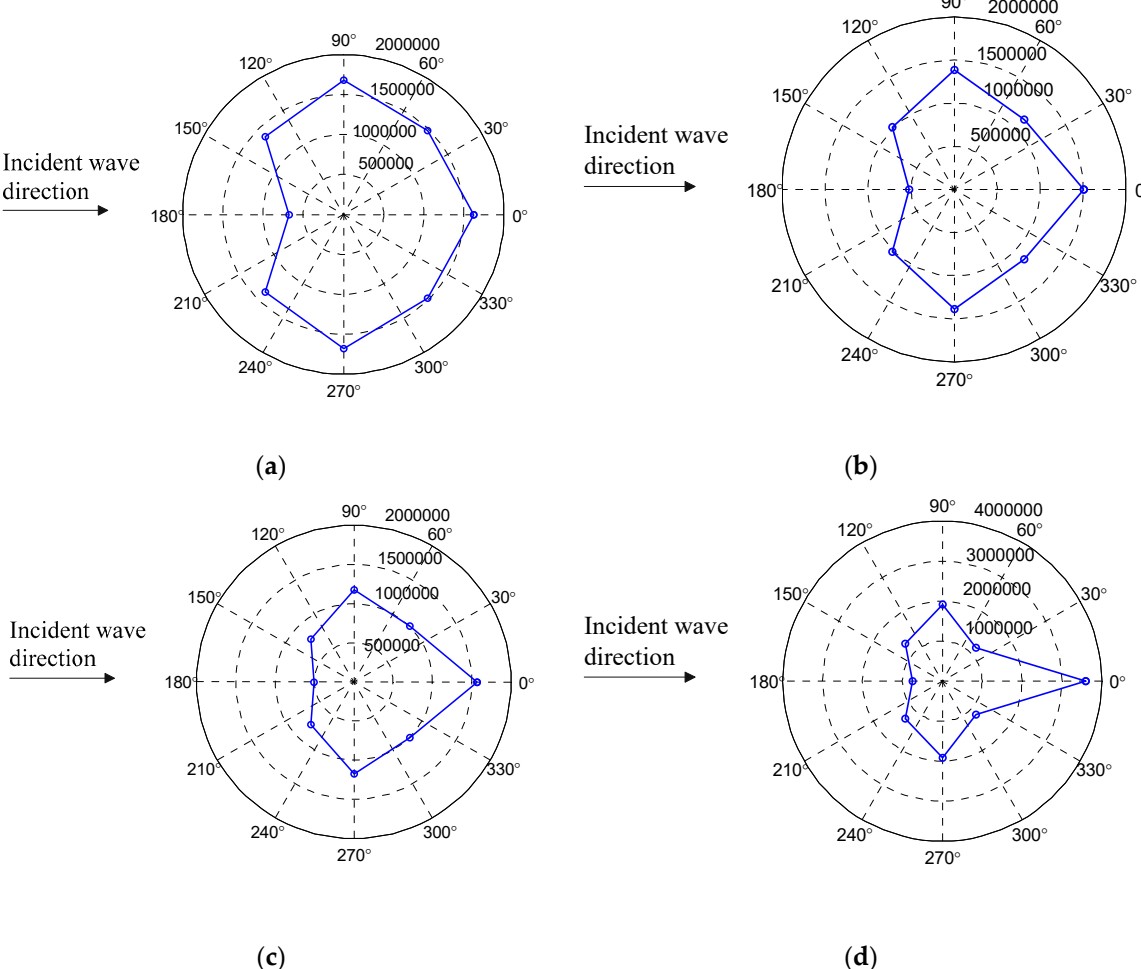

**Figure 12.** Direction patterns of nonlinear parameter *β* for (**a**) 1.4 mm; (**b**) 1.6 mm; (**c**) 1.8 mm; (**d**) 2.0 mm height micro-cracks with fixed 8 mm length.

The relative variations of the acoustic nonlinearity parameter $\beta'$ with micro-crack lengths ($l$ = 2, 4, 6, 8 mm) at fixed height ($h$ = 1.8 mm) for three kinds of excited mixed-frequency signal sources were investigated for the buried micro-crack with characteristic of zero width. The results are displayed in Figure 13a, which show that at the mixing frequencies ($f_1$ = 450 kHz, $f_2$ = 600 kHz) the magnitude of $\beta'$ monotonically increased with increasing micro-crack length, and at both mixing frequencies ($f_1$ = 470 kHz, $f_2$ = 650 kHz) and at ($f_1$ = 500 kHz, $f_2$ = 700 kHz) the magnitude of $\beta'$ first increased and then decreased with increasing micro-crack length which means that the forward scattering degree of the micro-crack increased with increasing buried micro-crack length at the initial stage of micro-cracking and then decreased with increasing length of micro-crack. These variation tendency means that the change property of acoustic nonlinearity parameter $\beta'$ for the buried crack is consistent to the through-thickness crack. Because of the buried micro-crack was embedded in different depths in the plate, the directivity of micro-crack could be carried out by changing the height of micro-crack (the buried depth of micro-crack). The variation of the relative acoustic nonlinearity parameter $\beta'$ for micro-cracks of fixed length ($l$ = 8 mm) and various heights ($h$ = 1.4, 1.6, 1.8, 2 mm) is displayed in Figure 13b, which show that in the case of three types of mixing frequencies the $\beta'$ monotonically increased with increasing micro-crack height and the tendency were apparent of a decreasing trend for the forward-scattering degree with increasing depth of buried micro-crack.

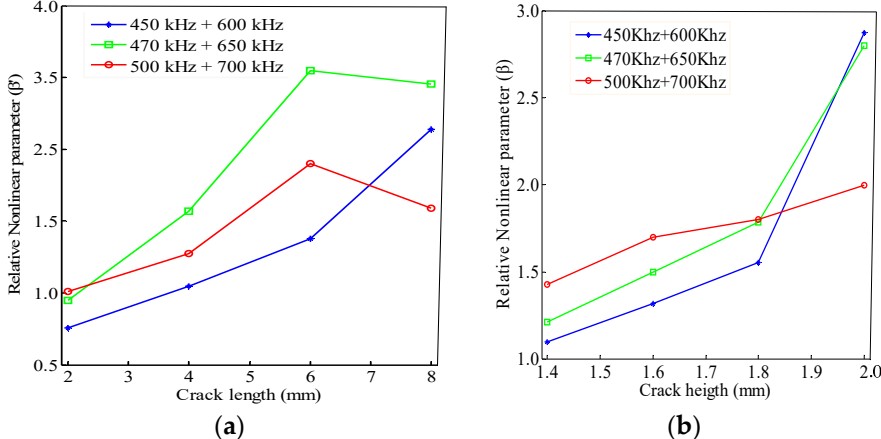

**Figure 13.** Relative nonlinear parameter $\beta'$ for (**a**) crack-length variation; (**b**) crack-height variation with three types of mixed-frequency incident signals.

## 4. Conclusions

In this research, nonlinear Lamb wave was used with frequency-mixing technique to analyze the scattering features of wave propagation at micro-cracks to characterize the directivity of these cracks. An analysis of the interaction of Lamb wave with micro-cracks was performed using finite element simulation. The pulse inversion method was used for effective extraction of the sideband signals at the sum or difference frequency components. The acoustic nonlinearity parameter $\beta$ which is related to the sideband signal at the difference frequency and the acoustic relative nonlinearity parameter $\beta'$ which reflects intuitively the scattering features of micro-cracks were introduced and proposed. Both the through-thickness micro-crack and the buried micro-crack were modulated in plate. The results show that:

(1) For the two kinds of micro-crack models, the value of $\beta$ in forward scattering has the largest magnitude, whereas $\beta$ shows its smallest value in the backward-scattering direction compared to other directions.

(2) For the two kinds of cracks the magnitude of $\beta'$ increased in the initial stage and then decreased with increasing micro-crack length, the reason is that as the micro-crack became longer, the contact stiffness of the interface firstly increased and then decreased.

(3) The variation tendency of $\beta'$ was decreased monotonically with increasing width of the through-thickness crack or with increasing burial depth of the buried crack, because in the both cases the contact stiffness of the crack interface decreased continuously.

**Author Contributions:** L.G. conceived of and designed the simulations. M.Z. analyzed the data and wrote the paper. X.W. investigated the simulations. Y.L. audited the content and reviewed the content. All authors have read and agreed to the published version of the manuscript.

**Funding:** This research was supported by the National Natural Science Foundation of China (grants 61571222 and 11774167), the Natural Science Foundation of Jiangsu Province, China (grant BK20161009), and the Six Talent Peaks Project of Jiangsu Province, China.

**Conflicts of Interest:** The authors declare no conflict of interest.

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
