# Peer review of "Nonlinear Lamb Wave Micro-Crack Direction Identification in Plates with Mixed-Frequency Technique"

_applsci, doi:10.3390/app10062135_

Round 1

Reviewer 1 Report

The topic of the study is interesting, as it considers the directionality of nonlinear effects in lamb waves dispersed on a crack. A weakness is that the study is purely numerical and the results might be specific to the considered setup: there is just an observation that an effect occurrs, but no attempt at its analysis.

My recommendation is a major revision.

GENERAL REMARKS:

This manuscript seems to build upon ref.23, which considered the same parameter, but not it directionality. This should be probably more explicitly mentioned in the introduction.

Please discuss the practical implications of the research, including:
1) potential for experimental verification and the implementation of the "wave-mixing technology"
2) potential for practical application
3) possible sensor patterns in pratical applications (they cannot encircle the crack, as one does not know, where the crack will appear)

Please correct the crucial sects.2.2-2.3. Now they rely on eq.1, which is unreadably formatted and seems to be taken out of the hat.

Several equations are unreadable. It seems to be a pdf coversion error, but it is authors role to verify the filesbefore submission. Please correct:
eq.1, first line of eq.3, most of eq.4, first lines of eqs.6 and 7, eq.12

TECHNICAL REMARKS:

sect.2.1: "Ultrasonic Lamb waves are stress waves generated by acoustic guided waves..."
Please correct.

sect.2.1: "where the plate thickness is of the same order of magnitude as the excitation signal wavelength [23]" is immediately followed by "Lamb waves can be generated when the thickness of the thin plate and the wavelength of the excitation signal are in the same order of magnitude [24]."
1) This is repetition of the the same information. Please correct.
2) What "wavelength of the excitation signal"? Please correct.

sect.2.1: "Lamb wave propagation has dispersive and multi-mode aspects, that is, the propagation velocity of Lamb waves is related not only to the elastic constant and material density, but also to the wave frequency and plate thickness."
The conjunction "that is" does not seem to be proper here: the 2nd part of the sentence is unrelated to the first part.

sect.2.1:"the displacement of the S0 mode is almost uniform in the thickness range of the plate"
It does not seem to be true. A0 mode is probaby meant here.

line 136: "using the fast fourier transform...". The FFT is a numerical procedure. Here, the authors operate on analytical formulas.

line 144: "When the experimental condition of the propagation distance and the wavenumber is identical,"
Please explain.

eq.5: what +/- does stand for? Is here + or - meant? If both, then how?

line 152: What is the "second harmonic contrast"?

eq.9: the negative exponents of e seem to be missing from the equation

sect.2.3: please explain first the physical idea behind the described technique. How is the excitation and then addition of the two signals shifted in phase realized in reality?

sect.3.1.1: the considered crack model is very simplistic

sect.3.2.2: what was the orientation of the crack?

Fig.9: there is almost no difference between subplots, at least practically, while the description emphasizes the minute differences.

Fig.10: the curves are very different for various frequencies. To what extent the results can be generalized to other frequency pairs? Which frequency pairs should be selected in practice?

Reviewer 2 Report

The article “Nonlinear Lamb wave micro-crack direction identification in plates with mixed-frequency technique” mainly deals with the phenomenology involved by the area of health monitoring and damage identification on technical systems. The authors proposed an innovative method, based on both the 3D finite element simulations, and the experimental investigations. They was shown that the acoustic nonlinearity parameter varied for different directions of the crack and it is possible to acquire useful information regarding the scattering features of the nonlinear Lamb wave interacting with the micro-crack. Thus, it becomes easier the characterization of the micro-crack directivity. Finally, it has to underline the results presented into this study, which enable and sustain the using of nonlinear Lamb wave, associated with frequency mixing technique, in order to identify the micro-cracks direction. From this perspective, this research is obviously significant. The article could be accepted for publication as it is.

Author Response

        Thanks for your review of this paper, and for your recognition of our research work. Based on this study, our research group will make further in-depth research and continue to make new progress. Once again, thank you very much for your comments.

Round 2

Reviewer 1 Report

The topic of the study is interesting, as it considers the directionality of nonlinear effects in lamb waves dispersed on a crack. A weakness is that the study is purely numerical and the results might be specific to the considered setup: there is just an observation that an effect occurrs, but no attempt at its analysis.

The authors have satisfactorily responded to most of my remarks. My current recommendation is a minor revision.

GENERAL REMARKS:

The pdf paper made available to this reviewer has still formatting errors in several equations (eq.1, first line of eq.3, most of eq.4, first lines of eqs.6 and 7, eq.12, etc.). There is no Word version available. This should be corrected in the final version.

Please discuss in the manuscript eq.1, which should be justified, discussed and explained. Currently, it seems to be taken out of the hat.

TECHNICAL REMARKS:

sect.2.1:"the displacement of the S0 mode is almost uniform in the thickness range of the plate"
Even though the authors confirm that S0 mode is meant here, I dare doubt it. Please verify: S0 is the symmetric mode, which has opposite displacements above and below the neutral axis. So, the displacement is not uniform.

line 167: "in second harmonic extract" should be probably "for extracting the second harmonic"

sect.3.2.2: Please state clearly in the manuscript, what is the orientation of the crack. Is it along the x axis? Or perhaps y axis?
